# Creatine Kinase and Myoglobin Plasma Levels in Mountain Bike and Road Cyclists 1 h after the Race

**DOI:** 10.3390/ijerph19159456

**Published:** 2022-08-02

**Authors:** Rafal Hebisz, Jacek Borkowski, Paulina Hebisz

**Affiliations:** Department of Physiology and Biochemistry, Wroclaw University of Health and Sport Sciences, 35 J.I. Paderewski Avenue, 51-612 Wroclaw, Poland; rafal.hebisz@awf.wroc.pl (R.H.); jacek.borkowski@awf.wroc.pl (J.B.)

**Keywords:** markers of muscle damage, cycling, competition

## Abstract

The aim of this study was to determine if 1 h after a cycling race, changes in plasma creatine kinase activity (CK) and myoglobin concentrations (MB) differ between mountain bike and road cyclists and if these changes show any correlation with race performance. Male mountain bike cyclists (*n* = 11) under 23 years old and male road cyclists (*n* = 14), also under 23 years old, were studied following one of their respective races. The cyclists had blood drawn 2 h before and 1 h after the race to assess CK and MB, then the change in pre- and post-race difference was calculated (ΔCK and ΔMB). Each cyclist’s performance time was recorded and the time difference from the winner was calculated (T_D_). The cyclists’ aerobic capacity was assessed during the incremental test, which determines maximal oxygen uptake and maximal aerobic power. It was observed that 1 h after the cycling race, CK (*p* = 0.001, *η*^2^ = 0.40, *F* = 15.6) and MB (*p* = 0.000, *η*^2^ = 0.43, *F* = 17.2) increased, compared to pre-race values. Post-race CK increased only in road cyclists, while post-race MB increased only in mountain bike cyclists. Smaller T_D_ were found for lower ΔMB in road cyclists but for higher ΔCK in mountain bike cyclists.

## 1. Introduction

Mountain bike (MTB) cycling in the Olympic format (XCO) is a highly intensive race lasting approximately 1 h 30 min, and is performed on technically demanding courses. During competition, the average heart rate exceeds 90% maximal heart rate [1,2], and power output measured during uphill climbs is frequently above maximal aerobic power, which is about 500 W for the elite level cyclists [1,3]. The average power measured during the MTB races was 283 W, also in elite level cyclists [1]. Biochemical analysis during simulated MTB racing showed that blood pH decreases to 7.3 and blood lactate concentration increases to 7.5 mmol/L [4]. Road cycling is of significantly longer duration and performed at a lower intensity. A classic one-day road race lasts from about 3 to even 7 h. Average power output at the elite level is approximately 200 W [5,6], with an average heart rate in the range of 50–60% maximal heart rate depending on the terrain [7].

Aside from differences in duration and exercise intensity, competitive road and MTB cycling also differ in muscle activity patterns. Existing literature has noted that the difficult terrain conditions of MTB cycling can elicit significant eccentric muscle contractions for shock attenuation and handling, particularly during downhill sections [2,3,4], and may compound energy expenditure even when little to no pedaling is required [4]. This type of varied and prolonged muscle action during intense cycling effort related to generating high power output can severely stress skeletal muscle [8] and cause structural damage within the muscle fiber [9], which had previously been confirmed in MTB cyclists [10]. The varied static, eccentric, and concentric phases can be observed in MTB [4], while in road cycling the concentric muscle action is definitely dominant [11].

The differences in road cycling and MTB racing formats (distance covered, terrain, and uphill/downhill phases) can also be reflected in pacing strategy, in which road cyclists show more linear and uniform pacing and power output [12] than MTB cyclists [1]. MTB cyclists are characterized by a greater involvement of fast twitch muscle fibers, which affects the generation of higher power output during climbs, compared to road cyclists specializing in classic single-stage races or multi-stage races [13]. The disparities in the physiological demands and activity profiles of these two competitive cycling modalities suggest that the skeletal muscle damage might take longer to occur in a road cycling race when compared to a MTB race. One of the causes of muscle damage could be cumulative oxidative stress [14]. Recent studies have used biochemical analysis to provide a more composite assessment of post-race oxidative damage [14]. It would seem prudent to continue this line of research to further assess and quantify post-race muscle damage in order to elucidate the physiological profiles of competitive road cycling and MTB. Research has found creatine kinase (CK) activity and myoglobin (MB) concentration to serve as useful markers of acute muscle damage and can be correlated with training status [8].

Creatine kinase is a dimeric globular protein consisting of two subunits with a molecular mass 43–82 kDa for each subunit. It buffers cellular ATP and ADP concentrations by catalyzing the reversible exchange of high-energy phosphate bonds between phosphocreatine and ADP produced during contraction [8,15,16]. Intense exercise that damages skeletal muscle cell structure at the level of sarcolemma and Z-disks results in an increase in total CK [15,16]. During intense exercise, the muscle tissue membrane permeability changes significantly and enzymes appear in the circulation [17]. Myoglobin is a monomer protein composed of 153 amino acids with a low molecular weight (18 kDa) [8,18]. There are three MB isoforms normally expressed in human muscle, and it is possible that MB has other roles in addition to oxygen storage and transport, including the regulation of nitric oxide at the microvascular and tissue level resulting in release of iron ions from the haem of myoglobin, which promotes the peroxidation of mitochondrial membranes [18,19]. Following strenuous exercise, MB is released as a result of the degradation of protein structures within muscle [8]. CK and MB as biomarkers of muscle status are particularly attractive, as previous studies have suggested that they can also be used to predict cycling performance [10].

Some studies show that blood CK activity is highest approximately 24 h after eccentric exercise [20] and after a long running effort of moderate intensity [21]. In other studies, the highest CK value was observed 8 h after intense strength training [22]. However, in the studies of Totsuka et al. [23], a statistically significant increase in CK activity occurred 3 h after exercise on a cycloergometer (the exercise lasted 90 min and was at low intensity). Park and Lee [24] showed that after downhill running, CK activity increased statistically significantly 2 h after exercise. Moreover, Koutedakis et al. [25] and Brancaccio et al. [8] showed that even 5 min after a maximal exercise test, CK activity increased significantly in rowers, whereas MB may increase within 30 min after exercise, with a peak value 1 h after exercise [8]. Therefore, it is worth wondering whether measuring CK activity and MB concentration in blood samples 1 h after mountain bike and road cyclists races will allow a determination of statistically significant differences between these groups. This is the time when all competitors are available at the race site, and after about 1 h they go back to their homes.

Understanding the relationship between potential changes in post-race CK activity and MB concentration and real-world race performance can have several practical implications. Previous studies have demonstrated a strong relationship between winning performance in road cycling and MTB and maximal oxygen uptake, power output at lactate threshold, and gross mechanical efficiency [26,27]. These variables reflect the aerobic energy potential of athletes and resistance to fatigue [9]. However, the onset of fatigue is modulated by a number of factors outside those which are cardiovascular, related to energy supply or the biomechanical, with current research highlighting the role of muscle trauma particularly during prolonged, high-intensity exercise [9]. A greater understanding of the relationship between the selected biochemical markers of post-race muscle damage and race performance could improve knowledge of the physiological determinants of success in MTB and road cycling.

The aim of this investigation was to, therefore, determine: (1) if 1 h after the cycling race there would be any significant changes in CK activity and MB concentration, (2) if the changes in CK activity and MB concentration differ between competitive road and MTB cyclists, and (3) if these biochemical markers show a relationship with race performance.

It was hypothesized that competition in the road and MTB race would result in greater CK activity and MB concentration 1 h after the race, and that the elevated levels of these biochemical markers would be negatively associated with race performance.

## 2. Materials and Methods

### 2.1. Participants

The study involved well-trained competitive road (*n* = 14) and MTB (*n* = 11) male cyclists. All participants were included in the same racing age category: under 23 (U23). No differences in baseline anthropometric measurements were observed between the two groups (Table 1).

The study design was approved by the institutional review board and conducted in accordance with the ethical standards established by the Declaration of Helsinki. Written informed consent was obtained from all participants and their guardians after the study details, procedures, and benefits and risks were explained.

### 2.2. Design and Procedures

Race data were collected during a National Cup event in which the cyclists competed in their respective discipline. Three days after race completion the aerobic capacity and performance of the cyclists was evaluated in controlled laboratory conditions at an exercise laboratory (PN-EN ISO 9001:2001 certified).

#### 2.2.1. Race Characteristics

The MTB cyclists competed in cross-country Olympic (XCO) race format with the start and finish located at 320 m above sea level. The MTB race consisted of six laps, each lap had a distance of 4.8 km and 180 m of elevation, and the total distance during the race was 28.8 km and 1080 m of elevation. The road cyclists competed in a one-day mass start race with the start and finish at 210 m above sea level. The road race consisted of eight laps, each lap had a distance of 17.5 km and 160 m of elevation, and the total distance during the race was 140 km and 1280 m of elevation. During races, both road and mountain bike cyclists drank isotonic carbohydrate drinks (containing maltodextrin, fructose and electrolytes) and water, and consumed carbohydrate energy gels (each gel containing 40 g of carbohydrates, including maltodextrin and fructose). Each cyclist drank about 200 mL of drink for each lap. During the entire race, MTB cyclists consumed two gels, while road cyclists consumed four gels (because their race was longer). The drinks were in bottles mounted on bicycles, while gels were in the pockets of cyclists shirts. The cyclists could refill their drinks and gels by driving through the buffet zone, marked on the race route. Both groups of cyclists provided the same type of carbohydrate drinks and energy gels. As demonstrated by Valentine et al. [28], depending on the type, carbohydrate and carbohydrate-protein drinks may have a different effect on indices of muscle disruption. The cyclists were familiarized with the course one day before competition by performing three (MTB) or two (road) practice laps. During the familiarization session, the participants were instructed to perform at an intensity below the second ventilatory threshold (VT2). The day preceding the familiarization session the cyclists rested and refrained from any exercise. Upon race completion, the official results (finish times) were posted and the difference between the winning time and each participant’s finish time (T_D_) was calculated as the time loss to race winner, and as a measure of race performance.

#### 2.2.2. CK and MB Determination

On the day of competition, a 0.5 mL sample of arterialized capillary blood from the fingertip was collected 2 h before race start and 1 h after race completion to determine CK activity and MB concentration. The blood samples were immediately centrifuged at 1000× *g* for 15 min at 4 °C for plasma recovery. These samples were stored in Eppendorf tubes at −30 °C and analyzed once all samples were collected. CK activity was determined with an EC 2.7.3.2 assay kit (Biosystems, Barcelona, Spain). MB concentration was determined using the Human Myoglobin Matched Antibody Pair Kit AB215407 (Abcam, Cambridge, UK) with the included protein standard. ExtrAvidin–Peroxidase conjugate (Sigma-Aldrich, Darmstadt, Germany) diluted to 1:1000 was added to the plate and incubated for 1 h at room temperature. The plate was then washed in phosphate-buffered saline containing 0.05% Tween 20. Afterwards, 0.4 mg/mL o-phenylenediamine and 0.3% H_2_O_2_ (*v*/*v*) dissolved in 0.1 M citrate buffer (pH 5.0) were added. The enzyme reaction was stopped after 30 min by adding 100 μL of a 1 M solution of H_2_SO_4_ and the absorbance was read at 490 nm (as the primary wavelength) using an Epoch TM spectrophotometer and integrated software (Bio-Tek Instruments, Winooski, VT, USA). The difference in pre- and post-race measures of CK and MB was then calculated (ΔCK and ΔMB, respectively). Creatine kinase activity and myoglobin concentration measured 1 h after the race were corrected for plasma volume changes. For this purpose, pre- and post-race hematocrit (HCT) value and hemoglobin (HGB) concentrations were also determined prior to centrifugation with an ABX Micros OT 16 Analyser (Horiba, Warsaw, Poland). These measures were used to assess the percent change in blood plasma volume (%ΔPV) in accordance with Dill and Costill [29]:$$\%\Delta \mathrm{PV}\;=\;100\;\times \;(\frac{\mathrm{HGBpre}}{\mathrm{HGBpost}})\;\times \;\left(\frac{1-\mathrm{HCTpost}}{1-\mathrm{HCTpre}}\right)\;-\;100$$

#### 2.2.3. Incremental Exercise Test

An incremental exercise test (IXT) was administered on an Excalibur Sport cycle ergometer (Lode B.V., Groningen, Netherlands) three days after race completion. The device was calibrated before each trial and the test began at a starting workload of 50 W that was increased every 3 min by 50 W until volitional exhaustion. If the participant was unable to complete a 3 min stage, 0.28 W was subtracted for each missing second from the obtained level of power to determine absolute and relative (per kg of body mass) maximal aerobic power (Pmax). Respiratory function was measured during the test with a face mask connected to a Quark gas analyzer (Cosmed, Rome, Italy). Calibration was performed prior to each trial with a reference gas mixture of carbon dioxide (5%), oxygen (16%), and nitrogen (79%). Tidal air was analyzed on a breath-by-breath basis to determine oxygen uptake (VO_2_), maximal oxygen uptake (VO_2_max), carbon dioxide excretion (VCO_2_), and minute pulmonary ventilation (VE). These measures were averaged over 30 s intervals. Absolute and relative VO_2_max were calculated based on VE and the composition of expired air. The second ventilatory threshold (VT2) was defined by V-slope analysis of VO_2_ and VCO_2_ [30] and absolute and relative power output at VT2 was determined.

### 2.3. Statistical Analysis

The Statistica 13.1 software package (StatSoft Inc., Tulsa, OK, USA) was used to analyze the data. The distribution of the data set was screened for normality using the Shapiro–Wilk test, and means ± standard deviations were calculated for each variable. Student’s *t*-test for independent samples was used to identify the between-group differences for the anthropometric and IXT-obtained measures. Differences in CK and MB between the groups and time points were tested using repeated-measures analysis of variance (ANOVA) with Duncan’s test post hoc when appropriate. Significance was accepted for all procedures at *p* < 0.05 and upper and lower confidence intervals (95% CI) are presented. The relationship between T_D_ and the biochemical and IXT-obtained variables was determined using multiple regression analysis. The explanatory variables were randomly included to develop the most optimal model. Only the regression models with the best fit and accepted at a significance level of *p* < 0.05 are presented.

## 3. Results

The IXT’s absolute and relative VO_2_max, absolute Pmax, and absolute and relative power output at VT2 were significantly greater, whereas the time loss to the race winner was significantly smaller in the group of road cyclists compared with the mountain bike cyclists (Table 2).

There was a significant main effect of time for CK activity (*F* = 15.6, *p* = 0.001, *η^2^* = 0.40) and MB concentration (*F* = 17.2, *p* = 0.000, *η^2^* = 0.43). A significant group × time interaction was observed for MB concentration (*F* = 4.5, *p* = 0.046, *η^2^* = 0.16). The results of Duncan’s test show a significant 1 h post-race increase in CK activity in the road cyclists and 1 h post-race increase in MB concentration in the MTB cyclists (Table 3).

Two regression models were selected that best explain the relative contribution of the different independent variables to the difference between winning time and finish time (the time loss to race winner) (Table 4).

Model 1 (*R* = 0.89, *R*^2^ = 0.79, *F* = 15.28, *p* = 0.002, *SEE* = 195.4) explains the time difference (T_1_) between MTB winning time and MTB cyclist’s finish time with the regression equation:T_1_ = 2822.7 − (525.15·VT2/kg) − (1.55·ΔCK)

Model 2 (*R* = 0.85, *R*^2^ = 0.73, *F* = 14.71, *p* = 0.001, *SEE* = 145.1) explains the time difference (T_2_) between the road race winning time and road cyclist’s finish time with the regression equation:T_2_ = 1758.53 − (4.18·Pmax) + (47.34·ΔMB)(1)

The 1 h post-race changes in CK and MB may be related to race performance, as high ΔCK in the MTB cyclists group or low ΔMB in the road cyclists group was achieved by cyclists who had a small time loss to the race winner.

## 4. Discussion

The presented study showed a significant increase in CK 1 h after the race only in the road cycling group. It is worth noting that the magnitude of change in CK was similar in both the road and MTB groups with similar effect sizes and that no significant between-group differences were found at either pre- or post-race. In turn, MB increased significantly 1 h after the race only in the MTB group. The lack of congruency in the direction of changes in CK activity and MB concentration may be conditioned by a variety of factors that should be mentioned.

A commonly used marker of exercise-induced muscle damage, CK activity is frequently assessed from 1 h to 72 h after exercise, and the peak is observed between 8–24 h after exercise [20,21,22,31]. MB concentration is assessed 1 h after exercise, which is when it reaches the peak [32]. The different temporal rates of change in CK activity and MB concentration may be caused by the difference in the molecular masses of CK and MB (43–82 kDa and 18 kDa, respectively), in which the greater mass of CK prolongs post-exercise clearance from muscle tissue to blood [33]. In the presented study, in the group of road cyclists the time from the race start to blood collection was about 4 h, and in the case of mountain bike cyclists it was about 2.5 h. In cycling it is typical that road races last longer than MTB races. The longer time from the race start to blood collection could have resulted in a statistically significant increase in CK activity among road cyclists, despite the fact that the size of CK changes was similar in both groups of cyclists. In the present study, blood samples were collected 1 h after the race due to the logistical considerations of the participants, who return to their place of residence after concluding the race. This early collection of post-race CK may not reflect actual post-exercise fatigue status. Nevertheless, it has been shown that 1 h after the cycling race, statistically significant changes in CK activity may occur. In addition, it is noteworthy that MTB cyclists had a higher CK level before the race than road cyclists (not statistically significant). This may be due to fact that on the day before the race the cyclists were driving along the race route to get to know its characteristics. Driving on the MTB route required more eccentric muscle work compared to road cyclists, which could have resulted in an increased level of CK.

For the analysis of the presented results, it seems important that the content of CK is two-fold greater in fast-twitch compared with slow-twitch muscle fibers [34,35]. Conversely, MB concentration is greater in slow-twitch fibers than in fast-twitch fibers; the ratio is about 1.5:1, respectively [36,37]. As a result, changes in CK levels may more accurately reflect fatigue status in fast-twitch muscle (which could have been similar in both the road and MTB groups), whereas the change in MB concentration may provide a better composite picture of slow-twitch fatigue (which was greater in the MTB group).

Although, CK activity did not change significantly after the MTB race, it was observed that ΔCK (which was calculated as the difference between the post-race and pre-race values) was a variable negatively related to the time difference between winning time and the participant’s finish time in Model 1 relating to MTB; cyclists with greatest ΔCK obtained the lowest T_D_. The increase in CK activity after exercise is ordinarily attributed to muscle cell membrane and tissue damage due to large muscle tension and eccentric exercise [8,31]. The intense ATP hydrolysis with a concurrent decrease in intracellular pH can lead to dysfunction of the Ca^2+^-ATPase pump, causing a large influx of calcium ions that can increase proteolytic enzyme activity. Such conditions can degrade cell membrane proteins, increasing cell membrane permeability and promoting an outflux of CK into the blood [31]. It is known that fast-twitch fibers are better adjusted to ATP synthesis via the glycolytic pathway than slow-twitch fibers [38]. Perhaps, the achievement of a small time loss to the race winner was conditioned by the ability to activate not only slow-twitch fibers but also, to a large extent, fast-twitch fibers.

The relationship between ΔCK and T_D_, shown in the Model 1 for MTB cyclists, could partly be attributed to the specificity of MTB racing, as it involves significant intensive eccentric exercise [2] that can result in significant muscle cell damage, particularly to Z-band muscle fibers [8,31]. The frequency of vibrations and the force of impact on unevenness along the race route could have been greater at higher cycling speeds, which could explain the relationship between ΔCK and the time difference between winning time and the participant’s finish time in the Model 1 for MTB cyclists.

Interestingly, the regression coefficient of ΔMB was positive (in contrast with ΔCK in the Model 1 for MTB cyclists) and an independent variable in the Model 2 predicted the difference between winning time and participant’s finish time in the road race. This suggests that it is race-induced damage to slow-twitch fibers that negatively affects winning time (increasing T_D_) in road cycling.

Finally, the developed regression models confirm the relative contribution of Pmax and power output at VT2 to performance in road and MTB cycling [6,26,27]. Borszcz et al. [39] observed a high correlation between mean power output in individual time trials and Pmax. Heuberger et al. [40] analyzed the predictive value of power at the lactate threshold with real-world road race performance and found moderate to strong correlations depending on the method used to calculate the lactate threshold. However, a study by Impellizzeri et al. [41] found MTB race performance to be moderately correlated with power output at VT2 normalized to body mass. Therefore, the presence of Pmax and VT2 in the models presented in this study is not surprising.

As a practical application, the present study shows that ΔCK in mountain bike cycling and ΔMB in road cycling can be used as parameters determining the physical fitness level of cyclists. ΔCK and ΔMB can complement the standard parameters, such as VO_2_max, Pmax, and power at the ventilatory threshold, in the assessment of physical fitness [6]. However, subsequent studies evaluating the effect of training processes (including eccentric endurance training such as downhill running or long downhill cycling on uneven ground) on the changes in ΔCK and ΔMB measured in cycling racing will be useful. Touron et al. [42] and Nakayama et al. [43] indicates that eccentric endurance training causes an acute increase in CK activity. On the other hand, Hurley et al. [22] observed that as a result of systematic strength training, the increase in CK and MB was smaller in the following weeks of the training process.

Study Limitations. The main limitation of the presented study was the requirement of taking biochemical measurements no later than 1 h after the race, because after this time, the cyclists returned to their homes. According to the previously cited literature, in the case of MB measurement, it was sufficient time to obtain the peak values. However, in the case of CK, it was certainly too short a time to measure the peak. The second limitation of the study was the collection of blood samples only once after the race, as it would be beneficial to collect a few blood samples during the first hour. The third limitation is the smaller size of the group of mountain bike cyclists (MTB cyclists: *n* = 11, and road cyclists: *n* = 14). As suggested by one of the Reviewers, this could have contributed to the lack of statistical significance for CK and to the conclusions that summarized the study. As CK in the mountain bike cyclists group changed from 320.1 to 392.3 [u∙L^−1^], it is a change by 72.2 [u∙L^−1^], but it did not reach the level of statistical significance. However, CK in the road cyclists group increased from 267.1 to 336.9 [u∙L^−1^], a change by 69.8 [u∙L^−1^], which achieved statistical significance.

## 5. Conclusions

The main findings of the study demonstrated a significant increase in CK activity 1 h after the race only in the road cycling group. In turn, MB concentration increased significantly 1 h after the race only in the MTB group. The 1 h post-race changes in CK and MB may be related to race performance, as high ΔCK or low ΔMB were obtained by better-performing cyclists (cyclists with small time loss to the race winner) in the MTB and road race, respectively.

## Figures and Tables

**Table 1 ijerph-19-09456-t001:** Anthropometric characteristics of the mountain bike and road cyclists.

	Mountain Bike Cyclists			Road Cyclists			Cohen’s *d*
	Mean ± SD	CI 95%		Mean ± SD	CI 95%		
		Upper	Lower		Upper	Lower	
Age [y]	19.1 ± 1.3	20.0	18.3	19.4 ± 1.6	20.4	18.5	0.21
BM [kg]	68.1 ± 5.5	71.8	64.4	71.1 ± 4.8	73.8	68.3	0.58
BH [m]	1.79 ± 0.04	1.81	1.76	1.77 ± 0.04	1.80	1.75	0.50
BMI	21.3 ± 1.2	22.2	20.5	22.6 ± 1.79	23.6	21.6	0.85

BM—body mass; BH—body height; BMI—body mass index; SD—standard deviation; CI—confidence interval.

**Table 2 ijerph-19-09456-t002:** Incremental exercise test and race characteristics of the mountain bike and road cyclists.

	Mountain Bike Cyclists			Road Cyclists			Cohen’s *d*
	Mean ± SD	CI 95%		Mean ± SD	CI 95%		
		Upper	Lower		Upper	Lower	
VO_2_max [L∙min^−1^]	4.28 ± 0.51	4.62	3.94	4.82 ± 0.33 *	5.01	4.63	1.26
VO_2_max [mL∙min^−1^∙kg^−1^]	62.9 ± 5.2	66.4	59.4	67.9 ± 4.4 *	70.4	65.4	1.04
Pmax [W]	378.2 ± 41.2	405.9	350.5	415 ± 27.5 *	430.9	399.1	1.05
Pmax [W∙kg^−1^]	5.56 ± 0.52	5.91	5.21	5.86 ± 0.51	6.16	5.56	0.58
VT2 [W]	264.5 ± 43.6	235.2	293.8	315.4 ± 40.5 *	338.8	291.9	1.21
VT2 [W∙kg^−1^]	3.89 ± 0.55	4.26	3.52	4.46 ± 0.67 *	4.85	4.07	0.93
T_D_ [s]	644.3 ± 383.6	902.0	386.5	204.9 ± 255.9 *	352.6	57.1	1.35

VO_2_max—maximal oxygen uptake; Pmax—maximal aerobic power; VT2—power output at the second ventilatory threshold; T_D_—difference between winning time and finish time—the time loss to race winner; SD—standard deviation; CI—confidence interval; *—between-group difference significant at *p* < 0.05.

**Table 3 ijerph-19-09456-t003:** Pre- and post-race CK activity and MB concentration and blood plasma volume.

	2 h Pre-Race			1 h Post-Race		
	Mean ± SD	CI 95%		Mean ± SD	CI 95%	
		Upper	Lower		Upper	Lower
Mountain bike cyclists						
CK [u∙L^−1^]	320.1 ± 182.0	442.4	197.8	392.3 ± 237.3	551.7	179.6
MB [ng∙mL^−1^]	21.0 ± 14.3	30.6	11.4	31.6 * ± 20.1	45.1	18.1
ΔPV [%]				−4.1 ± 9.3	2.2	−10.3
Road cyclists						
CK [u∙L^−1^]	267.1 ± 131.7	343.2	191.1	336.9 * ± 143.7	419.8	253.9
MB [ng∙mL^−1^]	17.8 ± 4.3	20.3	15.4	21.3 ± 3.9	23.6	19.0
ΔPV [%]				0.1 ± 13.9	8.1	−7.9

CK—plasma creatine kinase activity; MB—plasma myoglobin concentration; ΔPV—change in plasma volume; SD—standard deviation; CI—confidence interval; *—pre- and post-race difference significant at *p* < 0.05.

**Table 4 ijerph-19-09456-t004:** Partial correlation coefficients between the race time difference and independent variables among the two regression models.

	VO_2_max [L∙min^−1^]	VO_2_max [mL∙min^−1^∙kg^−1^]	ΔCK [u∙L^−1^]	ΔMB [ng∙mL^−1^]	Pmax [W∙kg^−1^]	Pmax [W]	VT2 [W]	VT2 [W∙kg^−1^]
T_1_	-	-	−0.68 *	-	-	-	-	−0.86 *
T_2_	-	-	-	0.76 *	-	−0.64 *	-	-

T_1_—time difference between MTB winning time and MTB cyclist’s finish time; T_2_—time difference between the road race winning time and road cyclist’s finish time; VO_2_max—maximal oxygen uptake; ΔCK—pre- and post-race difference in creatine kinase activity; ΔMB—pre- and post-race difference in myoglobin concentration; Pmax—maximal aerobic power; VT2—power output at the second ventilatory threshold; *—correlation significant at *p* < 0.05.

## Data Availability

The data presented in this study are available on request from the corresponding author.

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
