# Peer review of "Creatine Kinase and Myoglobin Plasma Levels in Mountain Bike and Road Cyclists 1 h after the Race"

_ijerph, 2022, doi:10.3390/ijerph19159456_

Round 1

Reviewer 1 Report

See attached.

Author Response

Reviewer 1

General

The current manuscript looks to compares changes in CK and MB concentrations following a mountain bike and road cycling race. The study measured CK and MB prior to the event and 1 hour after the races end. Participates also completed an incremental exercise test 3-days post race in a laboratory setting. The study found that MB concentrations were significantly altered in MTB racers, whereas CK levels were significantly altered for road cyclist. While the study was well performed, there are changes that should be made to improve clarity of the manuscript prior to publication.

- Thank you very much for your review and the valuable comments.

COMMENTS

General

The manuscript will need some minor English language formatting. There are issues with tense and grammar in some areas.

- After applying the corrections suggested by the Reviewers, we sent the manuscript to the Translator, who caught a few errors.

 Introduction

General

I feel like the introduction would benefit with more discussion about the importance of creatine kinase activity and blood plasma myoglobin concentration. Why are these important measures and what are they an indication of?

- As suggested, more information has been added about creatine kinase activity and myoglobin concentration, in the Introduction section of the fourth paragraph.

Page 2, Lines 43-44: I’m not show why this sentence is here staying that you can compare MTB with road cycling. More background is needed to clarify.

- Thank you for catching this mistake. The sentence has been corrected.

Page 2, Lines 64-67: Your sentence starting “Therefore, it is wondering whether…” seems to indicate a fatal fault in your research design. You should include research that has previously looked at levels of CK and MB at 1 hour post-exercise.

- The indicated fragment of the Introduction section has been changed. As suggested, we referred to studies showing significant changes in CK and MB during the first hour after exercise.

Methods

Page 3, Line 105-109: I am confused about the elevation gain. Is the stated elevation gain per lap or total? Please revise for clarifty.

- Indeed, it was not clearly written. Description has been changed:

"The MTB race consisted of 6 laps, each lap had a distance of 4.8 km and 180 m of elevation, the total distance during the race was 28,8 km and 1080 m of elevation."

"The road race consisted of 8 laps, each lap had a distance of 17.5 km and 160 m of elevation, the total distance during the race was 140 km and 1280 m of elevation."

Page 4, Line 146: You state you subtracted .28W for each second that was completed in a given stage. What is the rationale for doing this? Please include references that this is common practice or rational for why .28W was chosen.

- Each subsequent load during the incremental exercise test lasted 3 minutes (180 seconds), with the exception of the last load, which had a different duration as each participant ended the test when he was no longer able to continue it. During the test, the power increased by 50 W, so 50 W dividing by 180 seconds gives 0.28 W per second. So, if the participant was unable to exercise for the entire 3 minutes at the last test load, 0.28 W was subtracted from the finally obtained maximal power for each missed second.

Discussion

Your discussion needs a limitations section. You have several limitations within this study, the most prominent being that your window for testing CK and the window for peak elevation are not the same.

- Indeed, the window for testing CK is certainly the main limitation of these studies. As suggested, the study limitations have been added at the end of the Discussion section.

Reviewer 2 Report

Field type studies have weaker internal validity but just as importantly strengthen external validity.

I did not see any indication if there was or was not consumption of drinks during competition.  Given the potential effect of carbohydrate-protein drinks on indices of muscle disruption (RJ Valentine et al., 2008) could this have affected this studies results?

Changes are suggestions.

LINE

12-13    Change “Two groups participated in the 12 study: mountain-bike (n=11) and road cyclists (n=14). The study analyzed two races: mountain-bike 13 and road cycling race” to Mountain-bike cyclists (n=11) and road cyclists (n=14) were studied following one of their respective races.

14           Change “were” to had

16           Change to: Each cyclist’s performance time was recorded and the time difference from the winner was calculated (TD).

20-22     Change to: Smaller TDs were found for lower DMBs in mountain-bike cyclists but for higher DCKs in road cyclists.

49-50     Change “… should occur slower but may develop in a longer race time in road cycling relative to MTB” to … might take longer to occur in a road cycling race when compared to a MTB race.

90-94 including Table 1.                 My preference here for Participants is to describe what the target population is, then present the information of Table 1 at the beginning of the Methods section. But Journal & traditions may differ.

109        “The cyclists were instructed to provide a maximal effort with emphasis on obtaining a high finish.” Why give cyclists this instruction if they were in a competitive race?

171         Change “Obtained in incremental exercise test …” to The IXT’s …

176         I assume that the group main effect was non-significant for both CK and MB.

178         I assume that group x time interaction for CK was non-significant.

213         The differences between CK kinetics and MB is important, however I’m not convinced there is a difference between road racers and MTB racers for CK in this study because the group x time interaction (assumed from line 178 comment) indicates they both responded the same. The post hoc test may have found a significant difference for the road racers and not the MTB because the road racers had a couple of more participants.

263         I think the CK conclusion might be too strong. See 213 comments.

Author Response

Reviewer 2

Field type studies have weaker internal validity but just as importantly strengthen external validity.

- Thank you very much for your review and the valuable comments.

I did not see any indication if there was or was not consumption of drinks during competition.  Given the potential effect of carbohydrate-protein drinks on indices of muscle disruption (RJ Valentine et al., 2008) could this have affected this studies results?

- Indeed, no such information was provided. In the current version of the manuscript, we have added this information in the Materials and Methods section. During races, both road and mountain bike cyclists drank isotonic carbohydrate drinks (containing maltodextrin, fructose and electrolytes) and water, and consumed carbohydrate energy gels (each gel containing 40 grams of carbohydrates, including maltodextrin and fructose). Each cyclist drank about 200 ml of drink for each lap. During the entire race, MTB cyclists consumed 2 gels, while road cyclists consumed 4 gels (because their race was longer). Drinks are in bottles mounted on bicycles, while gels are in pockets of cyclists shirts. The cyclists can refill drinks and gels by driving through the buffet zone, marked on the race route. Both groups of cyclists provided the same type of carbohydrate drinks and energy gels, so it should not affect the results.

Changes are suggestions.

LINE

12-13    Change “Two groups participated in the study: mountain-bike (n=11) and road cyclists (n=14). The study analyzed two races: mountain-bike and road cycling race” to Mountain-bike cyclists (n=11) and road cyclists (n=14) were studied following one of their respective races.

- Thank you for your suggestion, part of the text has been changed.

14           Change “were” to had

- Done

16           Change to: Each cyclist’s performance time was recorded and the time difference from the winner was calculated (TD).

- The sentence has been changed.

20-22     Change to: Smaller TDs were found for lower DMBs in mountain-bike cyclists but for higher DCKs in road cyclists.

- The indicated sentence has been changed. I hope I have understood the suggestion properly. However, this sentence referred to the ΔMB and ΔCK, so I swapped the positions of mountains-bike and road cyclists: „Smaller TD were found for lower ΔMB in road cyclists but for higher ΔCK in mountain-bike cyclists.”

49-50     Change “… should occur slower but may develop in a longer race time in road cycling relative to MTB” to … might take longer to occur in a road cycling race when compared to a MTB race.

- The sentence has been changed.

90-94 including Table 1. My preference here for Participants is to describe what the target population is, then present the information of Table 1 at the beginning of the Methods section. But Journal & traditions may differ.

- I'm very sorry, but I don't understand this comment.

109        “The cyclists were instructed to provide a maximal effort with emphasis on obtaining a high finish.” Why give cyclists this instruction if they were in a competitive race?

- Thank you for catching this information, it is unnecessary and has been removed.

171         Change “Obtained in incremental exercise test …” to The IXT’s …

- Done

176         I assume that the group main effect was non-significant for both CK and MB.

- Yes, that's right.

The group effect was non-significant for CK (F = 0.64, p = 0.431, η2 = 0.027)

The group effect was non-significant for MB (F = 2.238, p = 0.148, η2 = 0.089)

178         I assume that group x time interaction for CK was non-significant.

- Yes, that is also right.

The group x time interaction for CK was non-significant (F = 0.005, p = 0.946, η2 = 0.0002)

213         The differences between CK kinetics and MB is important, however I’m not convinced there is a difference between road racers and MTB racers for CK in this study because the group x time interaction (assumed from line 178 comment) indicates they both responded the same. The post hoc test may have found a significant difference for the road racers and not the MTB because the road racers had a couple of more participants.

- With regard to the above comment and comments from other Reviewers, I raised the issue of the size of cyclists groups in relation to CK changes at the end of the Discussion section:

“The third limitation is the smaller size of the group of mountain bike cyclists (MTB cyclists: n = 11, and road cyclists: n = 14). As suggested by one of the Reviewers, this could have contributed to the lack of statistical significance for CK and to the conclusions that summarized the study. As CK in the mountain bike cyclists group changed from 320.1 to 392.3 [u∙l-1], it is a change by 72.2 [u∙l-1], but it did not reach the level of statistical significance. Whereas, CK in the road cyclists group increased from 267.1 to 336.9 [u∙l-1], it is a change by 69.8 [u∙l-1] which achieved statistical significance.”

263         I think the CK conclusion might be too strong. See 213 comments.

- Bearing in mind the comments of all Reviewers, I did not change the conclusions, but referred to this issue at the end of the discussion.

Reviewer 3 Report

The manuscript of Hebisz et al., entitled „Comparison of changes in creatine kinase activity and blood plasma myoglobin concentration 1 hour after the race in mountain bike and road cyclists, and the relationship of these changes with race performance” provides important new results in sports medicine. However, I have some minor concerns, that should be corrected:

1. The first sentence of the abstract seems to need grammatical revision („This study determine if…”)
2. It should be mentioned in the abstract, that only male cyclists, under 23 have been involved (because the result might be different in women or in different age group).

3. It should be mentioned in the abstract, that change of CK and MB means pre- and post-race.
4. Similarly, it should be also mentioned in the abstract that correlation between changes of CK and MB have been specific to one of the 2 groups, i.e. high ΔCK values in the MBT group or low ΔMB in the road cyclist group were achieved by cyclists who had a small time loss to the race winner. That should be mentioned among the results as well (e.g. just before Table 4), because actually it is mentioned only in the discussion.

5. In the introduction, the 2 types of cycling should be compared based on the same aspects, so if average power output is mentioned, it should be mentioned for both (if there are data about), or rather in case of none (if there are data only about one of them). It would be also important to mention in the comparison (since it is mentioned in the discussion as well), what is the role of short- or fast-twitch muscle fibers in the 2 types of cycling.

6. It could be also mentioned in the introduction, what is the connection between the 2 types of cycling and maximal oxygen uptake, etc. (parameters listed in Table 2), and why almost all of them are higher in the road cycling group (however, in the introduction it is mentioned that MBT is  frequently above maximal aerobic power).
7. It could be also interesting to write some sentences about in the discussion, that in the MBT group 2 hours pre-race CK and MB are tend to be higher than in road cyclists (however, it is not significant), with a possible explanation.

Author Response

Reviewer 3

The manuscript of Hebisz et al., entitled „Comparison of changes in creatine kinase activity and blood plasma myoglobin concentration 1 hour after the race in mountain bike and road cyclists, and the relationship of these changes with race performance” provides important new results in sports medicine. However, I have some minor concerns, that should be corrected:

- Thank you very much for your review and the valuable comments.

  1. The first sentence of the abstract seems to need grammatical revision („This study determine if…”)

- The sentence has been changed.

  1. It should be mentioned in the abstract, that only male cyclists, under 23 have been involved (because the result might be different in women or in different age group).

- As suggested, this information has been added.

  1. It should be mentioned in the abstract, that change of CK and MB means pre- and post-race.

- This information has been added.

  1. Similarly, it should be also mentioned in the abstract that correlation between changes of CK and MB have been specific to one of the 2 groups, i.e. high ΔCK values in the MBT group or low ΔMB in the road cyclist group were achieved by cyclists who had a small time loss to the race winner. That should be mentioned among the results as well (e.g. just before Table 4), because actually it is mentioned only in the discussion.

- Thank you for suggesting.

In the abstract, the indicated fragment has been changed, in line with the above suggestion and with the suggestion of another Reviewer: "Smaller TD were found for lower ΔMB in road cyclists but for higher ΔCK in mountain-bike cyclists."

The indicated information has also been added to the Results section: "The 1 hour post-race changes in CK and MB may be related to race performance, as high ΔCK in the MTB cyclists group  or low ΔMB in the road cyclists group were achieved by cyclists who had a small time loss to the race winner."

  1. In the introduction, the 2 types of cycling should be compared based on the same aspects, so if average power output is mentioned, it should be mentioned for both (if there are data about), or rather in case of none (if there are data only about one of them). It would be also important to mention in the comparison (since it is mentioned in the discussion as well), what is the role of short- or fast-twitch muscle fibers in the 2 types of cycling.

- Some information has been added in the first paragraph of the Introduction section to similarly describe mountain bike and road cycling. However, I have not found information indicating the average power of the MTB race, the unpublished data that I have from competitors winning the national championship indicate the value of approximately 300 W.

In the third paragraph of the Introduction section, information about the involvement and role of slow- and fast-twitch fibers in the two types of cycling has been added.

  1. It could be also mentioned in the introduction, what is the connection between the 2 types of cycling and maximal oxygen uptake, etc. (parameters listed in Table 2), and why almost all of them are higher in the road cycling group (however, in the introduction it is mentioned that MBT is  frequently above maximal aerobic power).

- Based on the available literature, cited in the presented manuscript, both mountain bike and road cyclists (racing at the world level) are characterized by a high oxygen uptake, above 70 ml∙min-1∙kg-1.

Higher values of the maxinal oxygen uptake, maximal aerobic power and the power at the ventilatory threshold in the group of road cyclists in comparison to mountain bike cyclists in the presented research are accidental, according to the literature data it is not a rule.

  1. It could be also interesting to write some sentences about in the discussion, that in the MBT group 2 hours pre-race CK and MB are tend to be higher than in road cyclists (however, it is not significant), with a possible explanation.

- In the second paragraph of the discussion, the following information was added:

"In addition, it is noteworthy that  MTB cyclists had a higher CK level before the race than road cyclists. This may be due to fact that on the day before the race the cyclists were driving along the race route to get to know its characteristics. Driving on the MTB route required more eccentric muscle work compared to road cyclists, which could have resulted in an increased level of CK."

Reviewer 4 Report

I read the manuscript entitled “Comparison of changes in creatine kinase activity and blood plasma myoglobin concentration 1 hour after race in mountain bike and road cyclist, and the relation of these changes with performance” which I found an interesting topic. However, I found some issues in the manuscript enumerated below to improve the quality of the manuscript before making a decision:

1.     Title needs to be adjusted as many variables confound future readers. Another title could be” Creatine Kinase and myoglobin plasma levels in mountain bike and road cyclist after 1-hour trial….. for example; I will eliminate the relation with performance.

2.     The rationale under the 1-hour blood measurements is not well established, as they mention the previous reports are three h in less time, but under low intensity and 24h after exercise (no mention the intensity); please explain.

3.     The authors mention in the introduction that intensity differences occur under these cycling modalities; more examples need to clarify how this could be an essential naming and load.

4.     The authors need to clarify the effect size range and calculations, as Cohen's d appears from nowhere, or is that supposed to be Duncan's post hoc test in table 2?

5.     In table 3, MB and CK levels are not different between MTB and road cyclist before the race? Can you please check and compare among groups, especially by the drop in plasma volume the observed post-race in MTB? Why do you think this happened?

6.     IN line 218, please place a citation on “Nevertheless, it has been shown that 1 hour after the cycling race, statistically significant changes in CK activity may occur”.

7.     Please clarify lines 238-240 as “It is, therefore, possible that ΔCK variability in mountain bike cyclists is dependent on the recruitment of fast-twitch fibers for ATP production to support slow-twitch fibers” if so, CK did not change; this stamen will be correct for MB. As MTB relies on type 2 fibers, one will be damaged more often. Please explain this.

8.     On the discussion, I am missing the applied focus on this, perhaps comment especially on performance training to modify variables. If not, this is simply a comparison and will be more valuable for future readers.

Author Response

Reviewer 4

I read the manuscript entitled “Comparison of changes in creatine kinase activity and blood plasma myoglobin concentration 1 hour after race in mountain bike and road cyclist, and the relation of these changes with performance” which I found an interesting topic. However, I found some issues in the manuscript enumerated below to improve the quality of the manuscript before making a decision:

- Thank you very much for your review and the valuable comments.

  1. Title needs to be adjusted as many variables confound future readers. Another title could be” Creatine Kinase and myoglobin plasma levels in mountain bike and road cyclist after 1-hour trial….. for example; I will eliminate the relation with performance.

- Thank you for the suggestion. The manuscript title has been changed.

  1. The rationale under the 1-hour blood measurements is not well established, as they mention the previous reports are three h in less time, but under low intensity and 24h after exercise (no mention the intensity); please explain.

- The indicated fragment of the Introduction section has been corrected with more details added. We cited additional studies showing a significant increase in CK a few minutes after maximal effort.

  1. The authors mention in the introduction that intensity differences occur under these cycling modalities; more examples need to clarify how this could be an essential naming and load.

- As suggested in the Introduction section, additional information is provided.

  1. The authors need to clarify the effect size range and calculations, as Cohen's d appears from nowhere, or is that supposed to be Duncan's post hoc test in table 2?

- Using the Student’s t test, it was calculated whether there were significant differences between the mountain bike and road cyclists groups in the assessed parameters presented in Table 1 and in Table 2. In these tables, Cohen's d serves as a measure of the effect size for the t-test.

Whereas,Table 3 shows the results of Duncan's post-hoc test and we incorrectly included Cohen's d in this table. Cohen's d values have been removed from Table 3.

  1. In table 3, MB and CK levels are not different between MTB and road cyclist before the race? Can you please check and compare among groups, especially by the drop in plasma volume the observed post-race in MTB? Why do you think this happened?

- We checked that the CK level did not differ statistically significantly between the MTB and road cyclists: p = 0.46 (before the race) and p = 0.44 (after the race).

We checked that the MB level did not differ statistically significantly between the MTB and road cyclists before the race: p = 0.52, while after the race the MB level was statistically significantly different between the MTB and road cyclists:  p = 0.04.

The lack of statistically significant differences between the groups of MTB and road cyclists may be related to the variability of CK and MB levels. In the case of MTB cyclists, the standard deviation for CK and MB exceeded 60% of the mean value. Perhaps it is influenced by the high variability of training and racing efforts among MTB cyclists (variable ground, variable slope of routes, variable intensity and rhythm of pedaling, variable nature of muscle work - alternately concentric on climbs and eccentric on down-hill and inequalities). For some cyclists, this may lead to the persistence of elevated CK values for several days. Moreover, on the day before the race the cyclists were driving along the race route to get to know its characteristics. Driving on the MTB route required more eccentric muscle work compared to road cyclists, which could have resulted in an increased level of CK. In the case of road cyclists, the values of standard deviation were much lower than among MTB cyclists.

Regarding the above comment and the suggestion of another Reviewer, a short note on this has been added in the second paragraph of the Discussion section.

  1. In line 218, please place a citation on “Nevertheless, it has been shown that 1 hour after the cycling race, statistically significant changes in CK activity may occur”.

- I'm very sorry, but I don't understand this comment. This citation in the indicated line is placed.

  1. Please clarify lines 238-240 as “It is, therefore, possible that ΔCK variability in mountain bike cyclists is dependent on the recruitment of fast-twitch fibers for ATP production to support slow-twitch fibers” if so, CK did not change; this stamen will be correct for MB. As MTB relies on type 2 fibers, one will be damaged more often. Please explain this.

- This sentence has been changed to: "Perhaps, the achievement of a small time loss to the race winner was conditioned by the ability to activate not only slow twitch fibers but also, to a large extent, fast twitch fibers."

  1. On the discussion, I am missing the applied focus on this, perhaps comment especially on performance training to modify variables. If not, this is simply a comparison and will be more valuable for future readers.

- I hope I understand the comment above. In response to this comment, I added the following text at the end of the Discussion section:

As a practical application, the presented study shows that ΔCK in mountain bike cycling and ΔMB in road cycling can be used as parameters determining the physical fitness level of cyclists. ΔCK and ΔMB can complement the standard parameters, such as VO2max, Pmax and power at the ventilatory threshold, in the assessment of physical fitness [6]. However, subsequent studies evaluating the effect of training process (including eccentric endurance training such as downhill running, long downhill cycling on uneven ground) on the changes in ΔCK and ΔMB measured in cycling racing, will be useful. Touron et al. 2021 and Nakayama et al. 2019 indicates that eccentric endurance training causes an acute increase in CK activity. On the other hand, Hurley et al. [15] observed that as a result of systematic strength training, the increase CK and MB was smaller in the following weeks of the training process.